# Quantized resistance revealed at the criticality of the quantum anomalous Hall phase transitions

Peng Deng [1] ✉, Peng Zhang[1], Christopher Eckberg [2,3,4], Su Kong Chong [1], Gen Yin [1], Eve Emmanouilidou[5], Xiaoyu Che[1], Ni Ni [5] & Kang L. Wang [1,5] ✉

In multilayered magnetic topological insulator structures, magnetization reversal processes can drive topological phase transitions between quantum anomalous Hall, axion insulator, and normal insulator states. Here we report an examination of the critical behavior of two such transitions: the quantum anomalous Hall to normal insulator (QAH-NI), and quantum anomalous Hall to axion insulator (QAH-AXI) transitions. By introducing a new analysis protocol wherein temperature dependent variations in the magnetic coercivity are accounted for, the critical behavior of the QAH-NI and QAH-AXI transitions are evaluated over a wide range of temperature and magnetic field. Despite the uniqueness of these different transitions, quantized longitudinal resistance and Hall conductance are observed at criticality in both cases. Furthermore, critical exponents were extracted for QAH-AXI transitions occurring at magnetization reversals of two different magnetic layers. The observation of consistent critical exponents and resistances in each case, independent of the magnetic layer details, demonstrates critical behaviors in quantum anomalous Hall transitions to be of electronic rather than magnetic origin. Our finding offers a new avenue for studies of phase transition and criticality in QAH insulators.

Topological phases of matter, such as quantum anomalous Hall (QAH)[1–7] insulators and axion insulators (AXI)[8–12], have attracted tremendous research interest in condensed matter physics. Due to their ferromagnetism and nontrivial band topology, when fully magnetized QAH insulators host dissipationless chiral edge states characterized by quantized Hall resistance and vanishing longitudinal resistance. When the magnetization is not uniformly polarized, however, otherwise quantized magnetic topological insulators may host alternative topological phases. Most notably, an even population of up and down magnetic domains may coalesce a normal insulator (NI) state, while an AXI phase is realized when uniformly oriented top and bottom surface magnetic orders are aligned antiparallel to one another. This latter state, which may be realized experimentally in molecular beam epitaxy

(MBE) grown magnetic topological insulator multilayers[10–12], has been the subject of scrutiny as a potential host for the topological magnetoelectric effect[8,9,13,14].

An applied magnetic field may be deployed to control magnetic topological insulator magnetic textures and, in so doing, drive transitions between NI, AXI, and QAH insulator electronic phases. During the passage to and from the QAH insulator state, a localization-delocalization electronic transition occurs[15], and, near criticality, fluctuations in the electronic localization length become sufficiently extended spatially and temporally that they dominate material transport properties. This dominance is evidenced by the frequent observation of universal scaling behaviors at the criticality[16–20]. Particularly, power-law evolutions of conductivity/resistivity dependent upon only

[1]Department of Electrical and Computer Engineering, University of California Los Angeles, Los Angeles, California 90095, USA. [2]Fibertek Inc, Herndon, VA 20783, USA. [3]US Army Research Laboratory, Adelphi, MD 20783, USA. [4]US Army Research Laboratory, Playa Vista, CA 20783, USA. [5]Department of Physics and Astronomy, University of California Los Angeles, Los Angeles, CA 90095, USA. ✉e-mail: dengpeng@g.ucla.edu; wang@ee.ucla.edu

a single parameter $L_s/\xi$, i.e., $\sigma_{\alpha\beta}(\rho_{\alpha\beta}) = f[(L_s/\xi)^{1/\nu}]$ are commonly reported in QAH materials[21–26]. Here $L_s$ is the inelastic scattering length that itself evolves as a power-law in temperature[27] $L_s \sim T^{-p/2}$, $\xi$ is the fluctuation correlation length which diverges at a critical point $x_c$ according to the relationship $\xi \sim |x - x_c|^{-\nu}$, $x$ is the tuning parameter driving the phase transition, and, finally, $\nu$ and $p$ are the correlation and temperature critical exponents. The critical behaviors derived from the above relationship are a product of the phase transition itself and, are nominally universal, depending only upon the fundamental symmetries and interactions present in the system. Consequently, they have historically been investigated as a powerful means to quantitatively categorize the fundamental details of the critical system. In QAH matter, however, the concurrence of magnetic and electronic phase transitions complicates the investigation of criticality as details of the magnetic system may pervade the electronic measurement.

Here we study critical behaviors of QAH insulators transitioning into both NI and AXI phases. As the magnetic details of the NI and AXI phases are drastically different, comparing critical behaviors across each material platform provides a route to unambiguously separate QAH signatures from those originating from the underlying magnetic ordering. To extract material critical properties, measurements are performed over a wide range of temperature and magnetic field. Data collected over the extended temperature range employed for this study was analyzed through traditional methodologies as well as a unique analysis protocol in which the temperature-dependent magnetic coercivity is accounted for. Although data treated using the former approach does not comply with finite-size scaling predictions, by deploying our new data handling technique we are able to accurately determine the critical points and critical values in these transitions. Strikingly, despite their considerable differences, we observe

consistent resistivity and conductivity tensor values at criticality in a QAH-NI and two separate QAH-AXI transitions. To further investigate the critical transport behavior, critical exponents of the QAH-AXI transition are extracted at two distinct magnetic reversal events. We report consistent critical exponents at both transitions, indicating that the critical behaviors in phase transitions of the QAH system are decisively electronic in origin.

## Results

The host platforms for the two distinct types of phase transitions explored in this study are both MBE-grown magnetic topological insulator thin films based upon the tetradymite topological insulator $(Bi, Sb)_2Te_3$. The QAH-NI transition is realized in 6 quintuple layers (QL) thick $(Bi, Sb)_2Te_3$ films uniformly doped with Cr ions. In these films, the QAH insulator becomes a normal insulator when the hybridization gap exceeds the magnetization gap, a condition that may be realized during magnetic reversals[15,28]. Fig. 1a shows a cartooned depiction of the domain configurations during the magnetization reversal process. When fully magnetized (Fig. 1a(i) and a(iii)), the system is a QAH insulator with a Chern number $C = 1$ or $-1$, exhibiting quantized $\sigma_{xy}$ and vanishing $\sigma_{xx}$ (Fig. 1b, c). When the sample enters a multi-domain state with equally populated up and down domains (Fig. 1a(ii)) the net magnetization is zero and the sample becomes a NI with $C = 0$ (see more discussion in Supplementary Information). The NI state is observed in the conductivity as a region where $\sigma_{xy}$ is equal to zero and $\sigma_{xx}$ is minimized, as can be seen in the magnetic hysteresis loop shown in Fig. 1b and c.

The AXI phase, meanwhile, is realized in trilayer structures composed of 3 QL Cr-doped $(Bi, Sb)_2Te_3$/6 QL $(Bi, Sb)_2Te_3$/3 QL V-doped $(Bi, Sb)_2Te_3$. In these films, V and Cr ions support magnetic ordering on

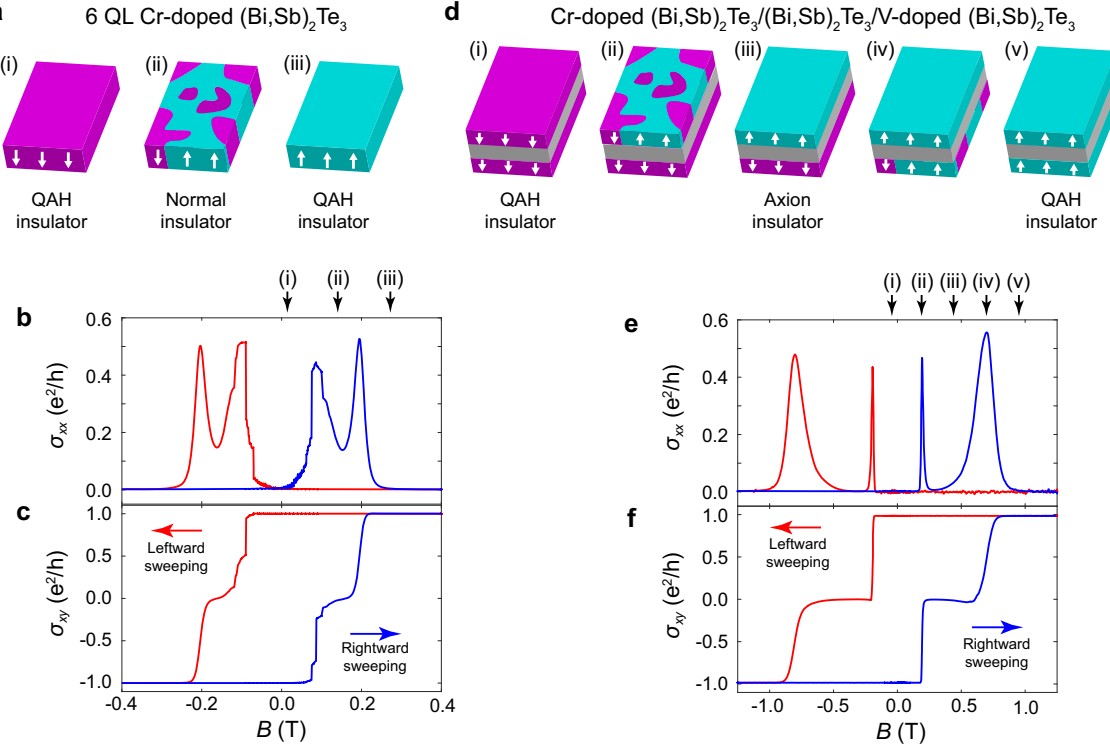

**Fig. 1 | Phase transitions in QAH insulators. a** Domain configurations during a QAH-NI phase transition. **b, c** Field dependences of $\sigma_{xx}$ and $\sigma_{xy}$ respectively for a 6 QL Cr-doped $(Bi, Sb)_2Te_3$ measured at 100 mK. The red and blue curves were acquired with the magnetic field sweeping to the left and to the right, respectively. The corresponding fields for (i)-(iii) states in (**a**) (during the rightward field sweeping) are highlighted by black arrows. **d** Domain configurations during a QAH-

AXI phase transition. **e, f** Field dependences of $\sigma_{xx}$ and $\sigma_{xy}$, respectively, for a 3 QL Cr-doped $(Bi, Sb)_2Te_3$/6 QL $(Bi, Sb)_2Te_3$/3 QL V-doped $(Bi, Sb)_2Te_3$ measured at 100 mK. The red and blue curves were acquired with the magnetic field sweeping to the left and to the right, respectively. The corresponding fields for (i)-(iv) states in (**d**) (during the rightward field sweeping) are highlighted by black arrows.

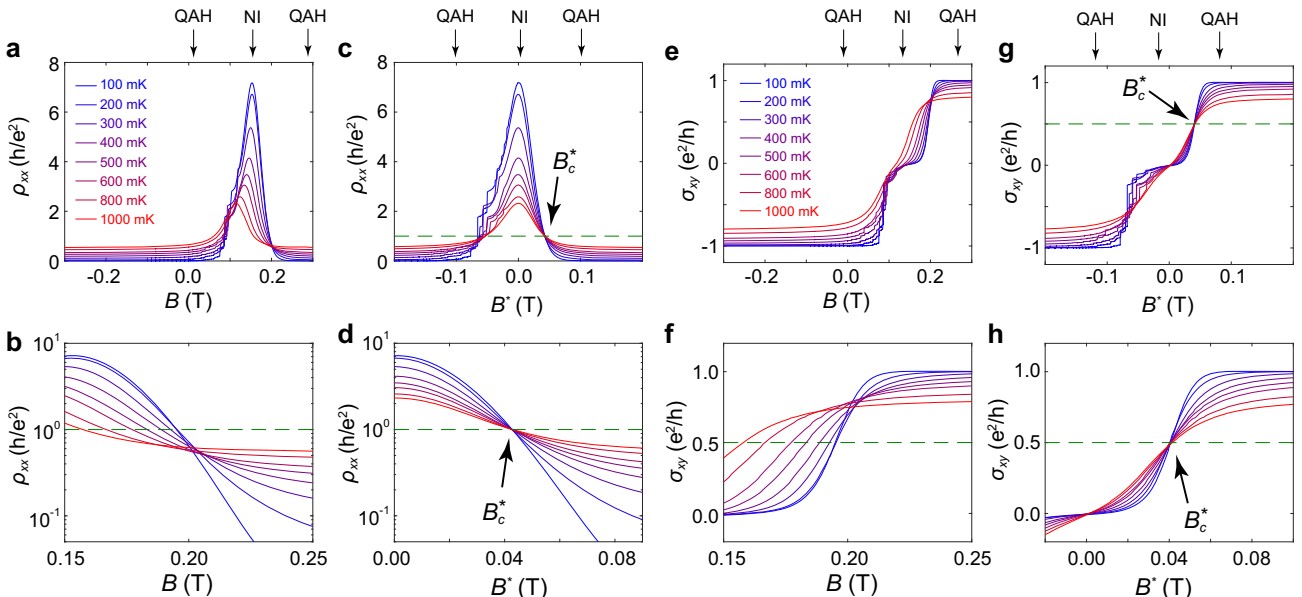

**Fig. 2 | Quantized $\rho_{xx}$ and $\sigma_{xy}$ revealed at the criticality of the QAH-NI transition.** **a** Field dependence of $\rho_{xx}$ under different temperatures. The field sweeping direction is from left to right. **b** Zoom-in plot of the Field dependence of $\rho_{xx}$. **c** $B^*$ dependence of $\rho_{xx}$ under different temperatures. All curves converge at $B^* = B_c^*$. **d** Zoom-in plot of the $B^*$ dependence of $\rho_{xx}$ around $B_c^*$. At the crossing point, the critical longitudinal resistance $\rho_{xx}^c = h/e^2$. **e** Field dependence of $\sigma_{xy}$ under different temperatures. The field sweeping direction is from left to right. **f** Zoom-in plot of the Field dependence of $\sigma_{xy}$. **g** $B^*$ dependence of $\sigma_{xy}$ under different temperatures. All curves converge at $B^* = B_c^*$. **h** Zoom-in plot of the $B^*$ dependence of $\sigma_{xy}$. At the crossing point, the critical Hall conductance resistance $\sigma_{xy}^c = 0.5\ e^2/h$.

the top and bottom surfaces respectively. When the magnetization in Cr-doped $(Bi, Sb)_2Te_3$ and V-doped $(Bi, Sb)_2Te_3$ layers are aligned parallel (Fig. 1d(i) and d(v)), the sample is in a QAH insulator state, showing the same vanishing $\sigma_{xx}$ and quantized $\sigma_{xy}$ (Fig. 1e, f) observed in the uniformly doped films. The V-doped layer features a much larger coercive field than the Cr-doped layer, such that magnetic reversal events at the top and bottom surfaces are staggered. Therefore, an antiparallel alignment occurs when transiting between the up and down magnetic state in these trilayered films. Under this antiparallel alignment (Fig. 1d(iii)), the system enters the AXI phase[10–12]. Although the AXI phase is born from a dramatically different magnetic state, it nonetheless produces the same transport features, i.e., zero $\sigma_{xy}$ plateaus and vanishing $\sigma_{xx}$, seen in the NI state (Fig. 1e, f).

In the following, we will present analyses of the critical behaviors of the QAH-AXI and QAH-NI quantum phase transitions, comparing the fundamental signatures of each and their dependence or independence upon magnetic property details. Particularly, we will analyze two observable effects predicted by finite-size scaling theory $\sigma_{\alpha\beta}(\rho_{\alpha\beta}) = f[(L_s/\xi)^{1/\upsilon}] = f[(x - x_c)T^{-p/2\upsilon}]$. First, we look for a temperature-independent conductance/resistance at the critical point $x = x_c$. Although several experimental and theoretical works report critical resistance values equal to the $h/e^2$ in quantum Hall systems[17,19], whether this value is universal or bears larger significance remains a subject of debate[29]. The second behavior we look for is a power-law temperature dependence of the tuning parameter derivative of the conductance/resistance when evaluated at the critical point $x_c$, i.e., $\partial\sigma_{\alpha\beta}(\rho_{\alpha\beta})/\partial x|_{x=x_c} \sim T^{-\kappa}$, where $\kappa = p/2\upsilon$. Given the significant role it plays in criticality analysis, we will note that the tuning parameter driving the localization-delocalization transition ($x$ in the above equations) in QAH systems is the material magnetization $M$[15]. This complicates identifying the critical point $x_c$ as $M$ cannot be addressed directly in experiments. Rather, it is instead controlled indirectly via the external magnetic field $B$. As such, $B$ is often treated as the phase transition tuning parameter in QAH criticality analyses, with this treatment being justified by the generally linear relationship between $M$ and $B$ near the critical point. Though $B$ is often approximately treated as the tuning parameter for phase transition in QAH materials,

it may lead to crucial errors in successfully identifying the location of the localization-delocalization critical point as will be discussed in the following.

We first consider the QAH insulator to normal insulator transition. Fig. 2a presents the field dependence of $\rho_{xx}$ under different temperatures with only the rightward sweeping data displayed for clarity (see leftward sweeping results in Supplementary Information). Note that in the plot, the $\rho_{xx}$ peak appearing during the magnetic switching is asymmetric with the inner edge displaying evident discrete jumps due to the Barkhausen effect[30]. To avoid erroneous conclusions in our analysis that may emerge from these discontinuities, we focus only on the outer edge of the transition as displayed in the zoom-in plot in Fig. 2b. It is immediately evident, however, that when plotted as a function of the magnetic field $B$, there is no unique value at which all of the $\rho_{xx}$ isotherms converge. This apparent deviation between the data and the scaling expectation is not due to a divergence between this system and the theoretical ideal, but rather due to the temperature-dependent magnetic coercivity in these films. As the magnetic reversal occurs at different magnetic fields at different temperatures, the applied field values at which $M = 0$ and $M = M_c$ occur will themselves have temperature dependencies. Though this temperature dependence may be ignored over a narrow temperature as has been done in previous works[4,21,23,26], it cannot be left unaccounted for in analysis conducted over a wide temperature range where the magnetic evolution may be consequential. This contrasts with quantum Hall systems[16,17], where the Landau level energy has a linear relationship with the external field value and no underlying temperature dependence.

To compensate for the effect of the temperature-dependent coercive field ($H_c$) and restore the true critical point, we conduct our analysis using an effective magnetic field $B^*$ defined as $B^* = B - \mu_0 H_c$, where the magnetic coercive field $H_c$ is identified as the location of the $\rho_{xx}$ peak[31]. Such a transformation guarantees the system has zero net magnetization when $B^* = 0$ at all temperatures, anchoring a good reference point. So long as $M$ maintains a linear relationship with $B$ during the magnetic reversal process, $M = M_c$ will occur at the same $B^* = B_c^*$ at all temperatures. To demonstrate the appropriateness of such

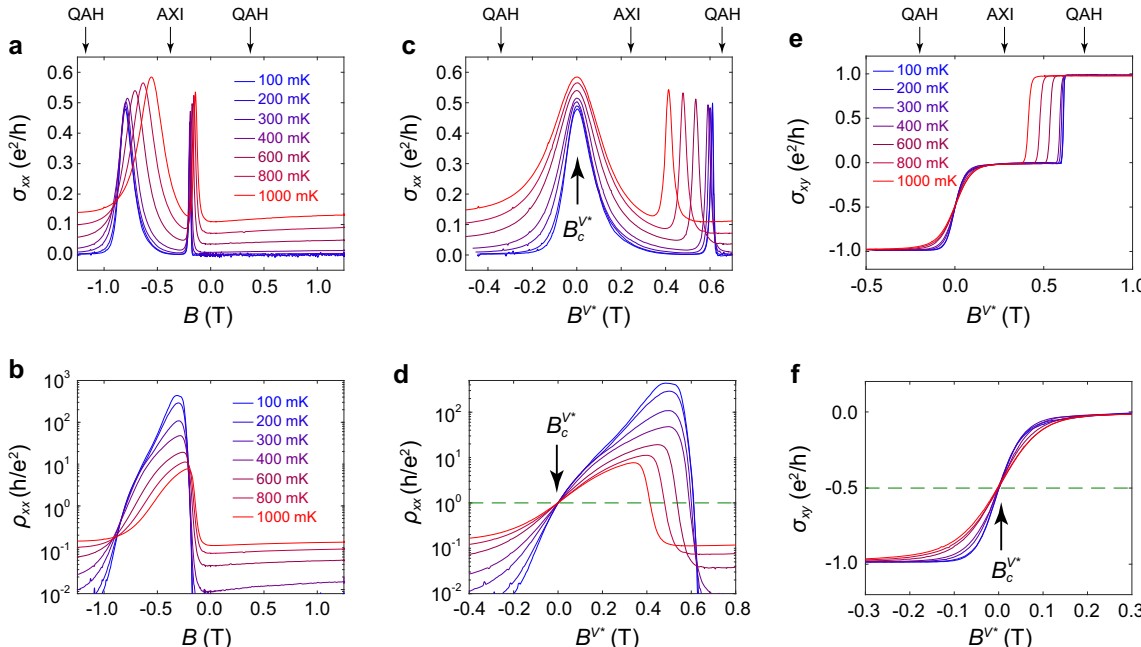

**Fig. 3 | QAH-AXI transition: V transition. a, b** Field dependences of $\sigma_{xx}$ and $\rho_{xx}$ under different temperatures, respectively. **c,** $B^{V*}$ dependence of $\sigma_{xx}$ under different temperatures, all broad $\sigma_{xx}$ peaks associated with V transition are aligned at $B^{V*} = 0$. **d** $B^{V*}$ dependence of $\rho_{xx}$ under different temperatures. All $\rho_{xx}$ curves cross at a single point $B_c^{V*}$ with a quantized resistance $\rho_{xx}^c = h/e^2$. **e** $B^{V*}$ dependences of $\sigma_{xy}$ under different temperatures. **f** A zoom-in plot of **e** at around $B_c^{V*}$ showing $\sigma_{xy}^c = 0.5\ e^2/h$.

a transformation, in Fig. 2c we present the $B^*$ dependence of $\rho_{xx}$. When plotted in this fashion, all curves converge on a single point $B_c^*$, confirming the veracity in this approach. Notably, the critical longitudinal resistance at $B_c^*$ is exactly one resistance quanta $\rho_{xx}^c = h/e^2$, which can be better recognized in the zoom-in plot shown in Fig. 2d. Similarly, when plotted against $B^*$, the Hall conductance also exhibits a quantized value of $0.5\ e^2/h$ at $B_c^*$, as shown in Fig. 2e–h.

We now discuss the QAH insulator to axion insulator transition in Cr-doped $(Bi, Sb)_2Te_3/(Bi, Sb)_2Te_3$/V-doped $(Bi, Sb)_2Te_3$ heterostructures. In these heterostructures, as opposed to the uniformly doped case, the magnetization reversal occurs as a "two-step" process, i.e., the magnetization in the Cr-doped $(Bi,Sb)_2Te_3$ layer flips first (this transition is referred to as Cr transition hereafter), while the V-doped $(Bi,Sb)_2Te_3$ layer flipping (referred to as V transition) occurs at a larger external field. Each reversal drives a QAH-AXI localization-delocalization transition. Fig. 3a shows several field dependent $\sigma_{xx}$ isotherms in an AXI sample as it is swept through its magnetic transitions. Here the broad and sharp $\sigma_{xx}$ peaks correspond to the V and Cr transitions, respectively. Fig. 3b shows the field dependence of $\rho_{xx}$ at different temperatures. Akin to the QAH-NI transition, the $\rho_{xx}$ curves do not converge during QAH-AXI transitions when the temperature-dependent coercivities of the different magnetic layers are left unaccounted for.

We again examine the conductivity as a function of an effective magnetic field accounting for the coercivity of the material. In the heterostructure, as the coercivity fields of V and Cr doped layers are widely separated, magnetization in one layer remains unchanged during the reversal of it in the other layer. Therefore, we can define two separate effective fields, $B^{V*} = B - \mu_0 H_c^V$ and $B^{C*} = B - \mu_0 H_c^C$, for the V and Cr transitions, respectively. Here, $H_c^V$ ($H_c^C$) is the field when the V(Cr)-doped $(Bi, Sb)_2Te_3$ layer has zero net magnetization, which is identified as the location of the corresponding $\sigma_{xx}$ peak[12]. Fig. 3c and d shows the $B^{V*}$ dependence of $\sigma_{xx}$ and $\rho_{xx}$ for different temperatures, respectively. Under such a shifting strategy, all broad $\sigma_{xx}$ peaks are aligned and a critical point is revealed at $B^{V*} = B_c^{V*}$ where all $\rho_{xx}$ curves converge. Like the uniformly doped $(Bi, Sb)_2Te_3$ discussed previously, a critical resistance of $\rho_{xx}^c = h/e^2$ is observed. Likewise, the $B^{V*}$ dependence of $\sigma_{xy}$ at different temperatures is shown in Fig. 3e and f. Again, all $\sigma_{xy}$ curves

cross at $B_c^{V*}$ with a critical value of $\sigma_{xy}^c = 0.5\ e^2/h$. An identical behavior is observed during the reversals of the Cr-doped layer as is shown in Fig. 4, although the reversal is much steeper as evidenced by the sharper $\sigma_{xx}$ peak.

The above results repeatedly show the same quantized transport coefficient values at criticality, indicating them to be universal signatures of QAH localization-delocalization phase transitions. To further confirm these results, we analyzed data collected under the opposite field sweeping direction in these same films (Supplementary Figs. 1 and 2), as well as transitions among different samples (Supplementary Fig. 3), all of which show the same $\rho_{xx}^c$ and $\sigma_{xy}^c$ values. The robust and fundamental nature of these signatures is emphasized by the drastically different electronic and magnetic properties of the NI and AXI samples. Notably, in the former, magnetic reversal processes are characterized by disordered magnetic domains scattered throughout the material, while in the latter the magnetization is uniformly oriented on one surface and disordered domains are confined to the other. Furthermore, within the AXI material, the critical conductance values are observed at two distinct magnetic reversal events of Cr and V doped layers, the uniqueness of which can be identified not only by their differing coercive fields, but also by the significant broadening at the transition of the V-doped layer compared with its Cr-doped counterpart. Finally, we will note that electronically the AXI and NI states are quite different, with the AXI phase displaying drastically more resistive behavior as evidenced by a $\rho_{xx}$ maximum two orders of magnitude larger than in the NI case.

The universal nature of the critical resistance/conductance implies an important characteristic of the QAH phase transition, i.e., the critical transport behaviors are purely electronic in origin and are not dictated by the magnetic criticality. To further elaborate on this point, we now turn to the other finite-size scaling feature commonly studied, i.e., the power law temperature dependence of $\partial \sigma_{xy}/\partial B$. The critical exponent $\kappa$ extracted from these experiments is nominally expected to be universal, and any changes in its values require a substantial material modification. On the other hand, reported values of $\kappa$ in QAH systems exhibit significant sample-to-sample variations[4,21–26,32], which may stem from the presence of disorders that interact over

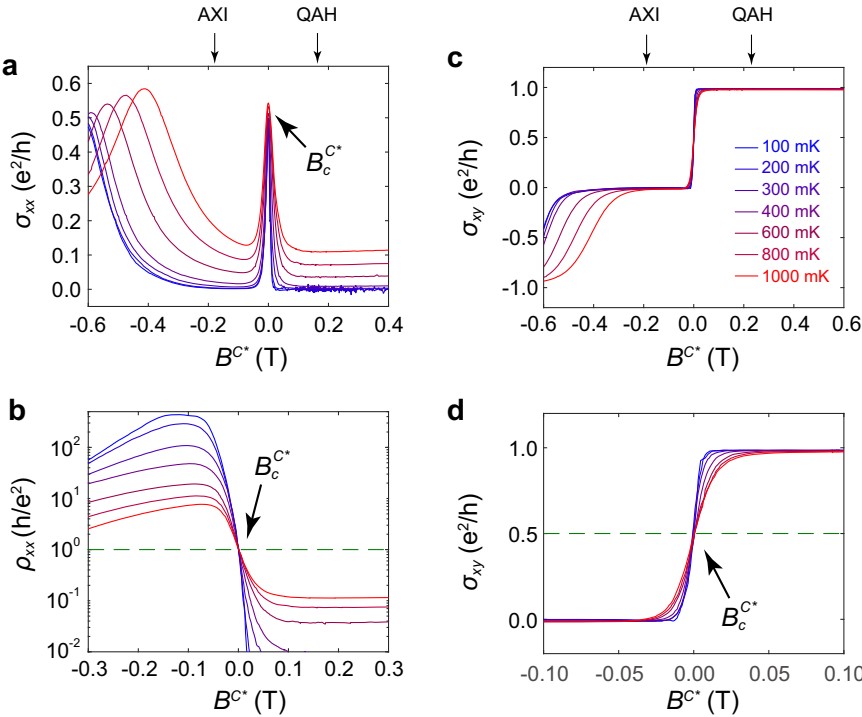

**Fig. 4 | QAH-AXI transition: Cr transition. a** $B^{C*}$ dependence of $\sigma_{xx}$ under different temperatures, all sharp $\sigma_{xx}$ peaks associated with Cr transition are aligned at $B^{C*} = 0$. **b** $B^{C*}$ dependence of $\rho_{xx}$ under different temperatures. All $\rho_{xx}$ curves cross at a single point $B_c^{C*}$ with a quantized resistance $\rho_{xx}^c = h/e^2$. **c** $B^{C*}$ dependences of $\sigma_{xy}$ under different temperatures. **d** A zoom-in plot of **c** at around $B_c^{C*}$ showing $\sigma_{xy}^c = 0.5\ e^2/h$.

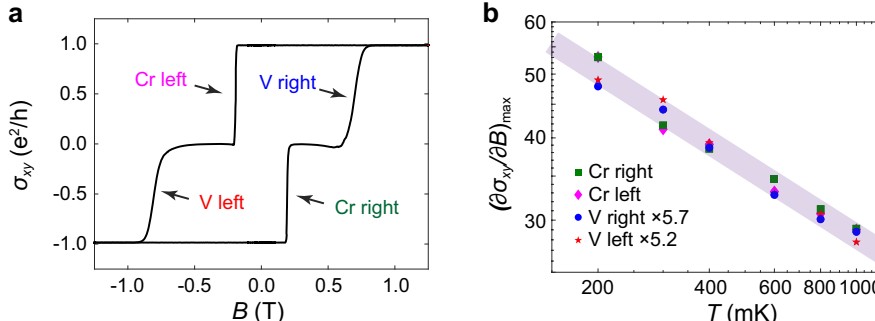

**Fig. 5 | Temperature of $(\partial\sigma_{xy}/\partial B)_{max}$ in the QAH-AXI transition. a** Field dependence of $\sigma_{xy}$. Four QAH-AXI transitions are indicated in the hysteresis loop. **b** The field derivative of the Hall conductance shows a power-law dependence on temperature as $(\partial\sigma_{xy}/\partial B)_{max} \sim T^{-\kappa}$ for all four transitions.

different length scales as has been previously reported in quantum Hall systems[33]. This introduces an additional uncontrolled parameter that complicates the comparison of critical behaviors among different samples with distinct magnetic structures, making it difficult to distinguish if the magnetic details of a given system exert any influence over the observed $\kappa$ value. Fortunately, AXI materials, which feature two unique magnetic structures embedded in a single system, provide an excellent testbed to extract the influence the magnetism exerts, if any, on the critical exponents.

Fig. 5 shows the temperature dependence of $(\partial\sigma_{xy}/\partial B)_{max}$ for four transitions in a hysteresis loop for one of our AXI devices. The four transitions correspond to V and Cr layer magnetic reversals under field sweeping along both the positive and negative directions. As can be seen in Fig. 5b, $(\partial\sigma_{xy}/\partial B)_{max}$ displays a linear relationship with $T$ in the log-log plot, indicating the expected power law $(\partial\sigma_{xy}/\partial B)_{max} \sim T^{-\kappa}$. While $(\partial\sigma_{xy}/\partial B)_{max}$ is about five times larger in Cr transition than in V transitions, indicating drastically divergent magnetic properties, the value of the critical exponent $\kappa$ [= (0.34, 0.36, 0.37, 0.36) ± 0.02 for the

Cr right, Cr left, V right, and V left transitions, respectively] is virtually the same for all transitions, reaffirming its electronic origins.

In summary, we present the critical behavior of localization-delocalization phase transitions of QAH insulators extracted via a novel criticality analysis methodology. By applying this data-handling approach to three different QAH transitions across two distinct material platforms we reaffirm the universal occurrence of quantized longitudinal resistance and Hall conductance at the localization-delocalization critical point across all samples measured. Further analysis in the AXI platform, meanwhile, indicates the critical exponent values are consistent across distinct magnetic reversal events; establishing that these exponents are purely electronic, and not a convolution of electronic and magnetic critical behaviors. Together, these findings provide repeated evidence that transport signatures at criticality in QAH insulators are robust against material variations and unaffected by magnetic details, while analysis methodologies developed for this work provide a route for expanded criticality investigations in QAH insulators in the future.

## Methods

The samples were prepared by molecular beam epitaxy growth in a Perkin-Elmer chamber with a base vacuum rate of $5 \times 10^{-10}$ Torr. Semi-insulating GaAs(111)B wafers were used as the substrate for the growth. High-purity Cr (99.995%), Bi (99.999%), Sb (99.999%), V (99.8%) and Te (99.9999%) were deposited on the epi-ready semi-insulating GaAs(111)B substrates. The epitaxial growth was monitored by an in situ RHEED, and the growth rate was calibrated by the RHEED intensity oscillation. For the Cr-doped $(Bi, Sb)_2Te_3$ sample, the Cr doping level is ~12%. In the Cr-doped $(Bi, Sb)_2Te_3/(Bi, Sb)_2Te_3/V$-doped $(Bi, Sb)_2Te_3$ hetero-structure, the Cr- and V-doping levels are about 15% and 10%, respectively. In both types of QAH samples, the Bi/Sb flux ratio is adjusted to tune the Fermi level into the magnetization gap to achieve quantization. The grown films were patterned into Hall bars with an effective size of 20 μm × 10 μm by photolithography and dry etching. The magneto-transport measurements were carried out on a Quantum Design Physical Property Measurement System with a dilution refrigerator insert.

## Data availability

The data that support the findings of the study are available from the corresponding author upon reasonable request.

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

## Acknowledgements

This work was supported by the NSF under Grants No. 1936383 and No. 2040737, the U.S. Army Research Office MURI program under Grants No. W911NF-20-2-0166 and No. W911NF-16-1-0472. C.E. is an employee of Fibertek, Inc. and performs in support of Contract No. W15P7T19D0038, Delivery Order W911-QX-20-F-0023. The views expressed are those of the authors and do not reflect the official policy or position of the Department of Defense or the US government. The identification of any commercial product or tradename does not imply endorsement or recommendation by Fibertek Inc. E.E. and N.N. were supported by the U.S. Department of Energy (DOE), Office of Science, Office of Basic Energy Sciences under Award Number DE-SC0021117.

## Author contributions

P.D. and K.L.W. conceived the project. P.D. and P.Z. performed sample growth and characterization. S.K.C. and X.C. fabricated the device. P.D., P.Z., C.E., and S.K.C performed transport measurements with the help of E.E. and N.N. P.D. and C.E. analyzed the data. G.Y. provided theoretical support. P.D., C.E., and K.L.W. wrote the manuscript with the inputs from all authors.

## Competing interests

The authors declare no competing interests.
