## [Peer Review File · Nature Communications]

REVIEWER COMMENTS

Reviewer #1 (Remarks to the Author):

The manuscript reports on the experimental investigation of the criticality at the transitions between the quantum anomalous Hall phase and the other insulating phases (normal insulator and axion insulator) in multilayered magnetic topological insulator structures. The transitions were driven by external magnetic field that affects both the electronic and magnetic properties of the structures. The authors designed a reliable methodology to disentangle the manifestations of the two types of effects in the critical behavior of the system across the topological phase transition. As a result, they observed universal properties of transport characteristics, as well as universality of critical exponents, which indicated the electronic nature of both transitions. The manuscript contributes with its important results to a very hot topic. The conclusions of the work are supported by the clever analysis of the data. I recommend the publication after the authors have made the following revisions to the manuscript related to their interpretation of the theoretical background.

1. The authors refer to a seminal work [19] where the quantized value of $\sigma_{xx}=0.5$ was predicted. However, there is still no theoretical consensus on this result for the mean conductance, and, moreover, on the meaning of this quantity. For a very recent discussion of this issue, please see PRL 129, 026801 (2022) and references therein, where the role of the mesoscopic fluctuations of σ_{xx} at the quantum Hall transition is emphasized. There is also a long record of other works published after Ref. [19] with conflicting conclusions.

2. The sentence "However, as κ is a composite of the critical exponents ν and p , it is sensitive to any material parameters that may impact either exponent, including sample disorders" should be refined. As it is stated now, the sentence diminishes the universality of the critical exponents in the scaling limit, and can be understood as if, by changing microscopic parameters of the structures, one can get arbitrary exponents. The point is that it is quite hard to change the universality class of the problem microscopically. Indeed, as the authors rightly mention, the scaling exponents are governed by the general symmetry of the model, as well as the global characteristics of interactions (long-range or screened short-range nature, for example). The present sentence can make an impression that the authors believe that the critical exponents may depend on such microscopic details as the screening radius, distance to the gates, etc., which is not the case. Of course, if the scaling limit is not achieved, the transient behavior of the system can be affected by such microscopic details, but in the thermodynamic limit the exponents are extremely robust to the change of the microscopic material properties.

Reviewer #2 (Remarks to the Author):

The manuscript presents electrical studies of two different material systems based on epitaxial heterostructures of topological insulator $(\text{Bi,Sb})_2\text{Te}_3$ with magnetic dopants. Such systems can display quantum anomalous Hall (QAH) insulator, normal insulator or axion insulator (AXI) phases and transitions between them. The data are presented for different magnetic fields values and temperatures between 100mK – 1K. Of particular interest is the critical behavior of the electrical signals. The values of the conductivity at the critical point are quantized. The analysis of the temperature dependence of the derivative of the transverse resistance on the magnetic field allows fitting the critical exponents. Authors claim that such critical behavior is of purely electronic but not magnetic origin, i.e. it depends only on the electronic structure and not on the magnetic behavior of the system.

The topic of the research is timely as it concerns modern, broadly discussed phases of matter such as QAH and AXI. Interesting observations are presented and conductivity values are indeed clearly quantized. General power law dependence in such structures has been already reported in X. Wu et al., *Nature Comm.*, 11, 4532 (2020). The novelty of this submitted manuscript seems to be in the analysis of the reorientation of the two magnetic layers within one structure that leads to the consistent values of critical exponents as well as the correction of the temperature dependence of the coercive fields. Unfortunately, I am not fully convinced about the careful and critical analysis of the acquired data. Below I present the list of the specific remarks:

1) Of major concern is the similarity between the datasets for Cr-doped $(\text{Bi,Sb})_2\text{Te}_3$ and Cr-doped $(\text{Bi,Sb})_2\text{Te}_3/(\text{Bi,Sb})_2\text{Te}_3/\text{V-doped } (\text{Bi,Sb})_2\text{Te}_3$ samples. Zero Hall plateau is present for the former sample. This is in contrast to M. Mogi et al., *Nature Materials* 16, 517 (2017) [Fig. 1F]. What is the reason for this zero Hall plateau? It can undermine one of the arguments for the attribution of the zero-Hall plateau to the axion insulator phase in the other sample. It can even undermine the proper attribution of conductivity features to the Cr and V doped layers reorientation.

2) Due to the above it can be speculated that the reorientation of the Cr-doped $(\text{Bi,Sb})_2\text{Te}_3$ seems to be a two-step process that is spread over a range of magnetic fields (approx. 0.1-0.2 T). Meanwhile, in the other sample, Cr reorientation is attributed to a very sharp conductivity change taking place in the narrow range of magnetic fields. I miss a comment clarifying the reason behind this.

3) Moreover, is it possible that Cr form inclusions or any kind of phase separation? Authors say that the layers are „uniformly doped” but can this be supported with any kind of post-growth characterization technique e.g. transmission electron microscopy (TEM)?

4) The coercive field dependence on the temperature is carefully accounted for. How about the temperature dependence of the range of fields over which the magnetic reorientation is spanned („width” of the reorientation in the magnetic field scale)? I miss a discussion about that and it seems that it can affect the values presented in Fig. 5b.

5) What are the Curie temperatures of the magnetic Cr and V doped layer? It seems that they shouldn't be close to the investigated region to rule out the magnetic critical behavior playing any role in the measurements.

6) After clarifying the origin of the separate transitions in Fig. 1b-c the similar discussion about the temperature dependence of the derivative of the transverse resistance on the magnetic field (Fig. 5b) for the Cr-doped only layer could be performed. This may add an important information to the discussion.

7) Data presentation in Fig. 5b is inappropriate - the span of the Y-axis should be adjusted to the data.

Due to the doubts and missing information as presented above I do not perceive the manuscript as suitable for Nature Communications.

Reviewer #3 (Remarks to the Author):

In the manuscript "Quantized resistance revealed at the criticality of the quantum anomalous Hall phase transitions", by Peng Deng et al., critical behavior of topological phase transitions, in particular the quantum anomalous Hall to normal insulator (QAH-NI), and quantum anomalous

Hall to axion insulator (QAH-AXI) transitions are investigated. Through their new data analysis protocol, the critical behavior of the QAH-NI and QAH-AXI transitions are evaluated over a wide range of temperature and magnetic field. One of the main outcomes of their analysis is the observation that critical behaviors in quantum anomalous Hall transitions is due to electronic rather than magnetic origin.

The manuscript is clear and well written; and also, the scope of this work fits well in the frame of Nature communications. It will attract the interests of readers in the physical sciences community. I have few comments that need to be clarified before its final acceptance:

1. As far as I can read in the main-text and methods, the Cr- and V-doping levels in their heterostructures is not indicated? To which extent the Cr- and V-doping level would influence the critical behaviors of the QAH-NI and QAH-AXI transitions?

2. Given the strong statement made that “Together, these findings provide repeated evidence that transport signatures at criticality in QAH insulators are robust against material variations.....” I have a question regarding “reproducibility” of these data: it is not clarified in the manuscript how many Hall bars/samples were measured for each particular types of two distinct material platforms illustrated in Fig. 1(a) and Fig.1(d). If more than one samples/Hall bar devices were measured and analyzed using the proposed data analysis protocol, authors should make clarifications on this, and it would be good to have this included in the manuscript. This will strengthen their claim and also help readers to understand how robust their new analysis protocol is.

3. It would be good to clarify at which temperatures the spectra in Fig. 1(b)-(f) were acquired. I understand it is probably at 100 mK for the blue and 1000 mK for the red one?

Response letter

Reviewer #1:

The manuscript reports on the experimental investigation of the criticality at the transitions between the quantum anomalous Hall phase and the other insulating phases (normal insulator and axion insulator) in multilayered magnetic topological insulator structures. The transitions were driven by external magnetic field that affects both the electronic and magnetic properties of the structures. The authors designed a reliable methodology to disentangle the manifestations of the two types of effects in the critical behavior of the system across the topological phase transition. As a result, they observed universal properties of transport characteristics, as well as universality of critical exponents, which indicated the electronic nature of both transitions. The manuscript contributes with its important results to a very hot topic. The conclusions of the work are supported by the clever analysis of the data. I recommend the publication after the authors have made the following revisions to the manuscript related to their interpretation of the theoretical background.

Response: We thank the reviewer for the nice summary and the positive comments of our work. We are very grateful to the reviewer for the recommendation for publication. Below please find a point-by-point response to the reviewer's questions.

1. The authors refer to a seminal work [19] where the quantized value of $\sigma_{xx}=0.5$ was predicted. However, there is still no theoretical consensus on this result for the mean conductance, and, moreover, on the meaning of this quantity. For a very recent discussion of this issue, please see PRL 129, 026801 (2022) and references therein, where the role of the mesoscopic fluctuations of σ_{xx} at the quantum Hall transition is emphasized. There is also a long record of other works published after Ref. [19] with conflicting conclusions.

Response: We thank the reviewer for pointing out the theoretical works regarding the critical value of σ_{xx} . We agree with the reviewer that the reference we previously cited does not fully represent the complexity of the issue. In light of the reviewer's comment, we have rewritten the corresponding sentences and changed the references (see #3 in the summary of changes below).

2. The sentence "However, as κ is a composite of the critical exponents ν and p , it is sensitive to any material parameters that may impact either exponent, including sample disorders" should be refined. As it

is stated now, the sentence diminishes the universality of the critical exponents in the scaling limit, and can be understood as if, by changing microscopic parameters of the structures, one can get arbitrary exponents. The point is that it is quite hard to change the universality class of the problem microscopically. Indeed, as the authors rightly mention, the scaling exponents are governed by the general symmetry of the model, as well as the global characteristics of interactions (long-range or screened short-range nature, for example). The present sentence can make an impression that the authors believe that the critical exponents may depend on such microscopic details as the screening radius, distance to the gates, etc., which is not the case. Of course, if the scaling limit is not achieved, the transient behavior of the system can be affected by such microscopic details, but in the thermodynamic limit the exponents are extremely robust to the change of the microscopic material properties.

Response: We thank the reviewer for this important comment regarding the universality of the critical exponents in the quantum phase transition. We certainly do not mean to imply that the critical exponents are arbitrarily valued and vary according to the microscopic details of the system. Rather, we simply intend to note that inconsistent κ values are reported in the literature, and this variability may complicate the direct comparison of scaling results across different reports, which inspired our current study where multiple magnetic transitions are explored in a single sample with guaranteed consistency in symmetry, interaction length scale, disorder type/level, etc.

We regret any confusion that may have been caused by our previous statement. To clarify this point we have rewritten the paragraph containing this sentence, taking care not to undercut the notion of universal critical exponents (see #4 in the summary of changes).

Reviewer #2:

The manuscript presents electrical studies of two different material systems based on epitaxial heterostructures of topological insulator $(\text{Bi,Sb})_2\text{Te}_3$ with magnetic dopants. Such systems can display quantum anomalous Hall (QAH) insulator, normal insulator or axion insulator (AXI) phases and transitions between them. The data are presented for different magnetic fields values and temperatures between 100mK – 1K. Of particular interest is the critical behavior of the electrical signals. The values of the conductivity at the critical point are quantized. The analysis of the temperature dependence of the derivative of the transverse resistance on the magnetic field allows fitting the critical exponents. Authors

claim that such critical behavior is of purely electronic but not magnetic origin, i.e. it depends only on the electronic structure and not on the magnetic behavior of the system.

The topic of the research is timely as it concerns modern, broadly discussed phases of matter such as QAH and AXI. Interesting observations are presented and conductivity values are indeed clearly quantized. General power law dependence in such structures has been already reported in X. Wu et al., Nature Comm., 11, 4532 (2020). The novelty of this submitted manuscript seems to be in the analysis of the reorientation of the two magnetic layers within one structure that leads to the consistent values of critical exponents as well as the correction of the temperature dependence of the coercive fields. Unfortunately, I am not fully convinced about the careful and critical analysis of the acquired data. Below I present the list of the specific remarks:

Response: We thank the reviewer for the nice summary of our work, and his/her recognition of current interest in QAH and AXI physics. We also appreciate the referee's correct assertion that the analysis in our present work is one of the several factors distinguishing it from the prior scaling studies in magnetic topological matter, including the Nat. Commun. paper by Wu and collaborators. Crucially, through this analysis we observe scaling phenomena that extends beyond what has been previously reported in literature as highlighted in the following:

First, in studying the criticality at both transitions in AXI films we observe consistent exponents at two distinct switching events. This consistency provides direct experimental evidence that the critical exponents observed in these materials is not influenced by the magnetic phase transition. Secondly, through this analysis we have demonstrated it is possible to greatly expand the temperature range over which critical scaling in magnetic topological insulators may be studied. For example, in the work of Wu et al., scaling is studied over a very narrow temperature range from 45 mK to 80 mK, whereas, we observe a robust scaling relation over a 900 mK window between 100mK and 1000mK. Finally, as Reviewer 1 has indicated, there exists some debate and controversy over the expected critical resistance of a quantum anomalous Hall material. Indeed, in the work of Wu et al., inconsistent values of $\rho_{xx}^c = 1.0 h/e^2$, $1.1 h/e^2$, and $2.1 h/e^2$ are reported (for three sample discussed in main text and supplementary). In the present study, meanwhile, consistent critical resistances of h/e^2 are observed over 12 QAH-NI or QAH-AXI transitions; a conclusion that may be missed if the temperature dependent magnetic coercive fields are left unaccounted for.

We additionally thank the referee for their questions regarding the rigor of our analysis protocol. Please refer below for a point-by-point response to the questions posed.

1) Of major concern is the similarity between the datasets for Cr-doped $(\text{Bi,Sb})_2\text{Te}_3$ and Cr-doped $(\text{Bi,Sb})_2\text{Te}_3/(\text{Bi,Sb})_2\text{Te}_3/\text{V-doped } (\text{Bi,Sb})_2\text{Te}_3$ samples. Zero Hall plateau is present for the former sample. This is in contrast to M. Mogi et al., Nature Materials 16, 517 (2017) [Fig. 1F]. What is the reason for this zero Hall plateau? It can undermine one of the arguments for the attribution of the zero-Hall plateau to the axion insulator phase in the other sample. It can even undermine the proper attribution of conductivity features to the Cr and V doped layers reorientation.

Response: We thank the reviewer for their important comment regarding zero Hall plateaus in AXI and QAH materials. In ultrathin TI materials, top and bottom surface states hybridize, coalescing a trivial hybridization gap m at Γ point. Additionally, the magnetic exchange gap m_0 is proportional to the magnetization and vanishes when up and down domains are equally populated. Consequently, during magnetization reversal events in ultrathin QAH materials the system will traverse through a normal insulator (NI) phase when $m_0 < m$. The zero Hall plateau in Cr-doped $(\text{Bi, Sb})_2\text{Te}_3$ originates from this NI phase and does not reflect a plateau in the sample magnetization [Phys. Rev. B 89, 085106 (2014); Phys. Rev. Lett. 113, 137201 (2014)].

With this in mind, the discrepancy between the Cr-doped $(\text{Bi, Sb})_2\text{Te}_3$ films reported in the present work and those seen in Fig. 1F of [M. Mogi et al., Nature Materials 16, 517 (2017)] can be accounted for in the thickness variations between the two studies. Mogi and coauthors report results for a 10 QL sample, while in the present study we report on 6 QL thick films which feature considerably stronger surface hybridization. Due to the comparatively weak hybridization in the 10 QL film studied by Mogi et al., their films do not enter a proper insulating state during magnetization reversals, and the conductance plateau is not present.

The vanishing of the insulating state at thicknesses beyond 6 QL is also observed in films grown in our lab, as we have previously reported in Fig. 1 in [Sci. Adv. 6, eaaz3595 (2020)]. To further confirm this point, we have recently grown an 8 QL thick Cr-doped $(\text{Bi, Sb})_2\text{Te}_3$ and the result is shown in Fig. R1. As can be clearly seen, no zero Hall plateau is observed in the 8 QL sample, in sharp contrast to the 6 QL one.

Figure R1. Field dependence of Hall conductance for an 8 QL thick Cr-doped $(\text{Bi, Sb})_2\text{Te}_3$ sample.

[Redacted]

Finally, although we rely on transport measurement to extract information regarding the magnetic behavior, combined measurement of transport and magnetic force microscopy [Phys. Rev. Lett. 120, 056801 (2018)] have previously been reported by other groups, confirming the independent switching of the top and bottom magnetic layers in the Cr-(Bi, Sb)₂Te₃/(Bi, Sb)₂Te₃/V-doped (Bi, Sb)₂Te₃ sample.

In the updated version, we have included more discussion of the zero Hall plateaus in Supplementary Information and added Fig. R1 as Supplementary Fig. 8 (see #12 in the summary of changes below).

2) Due to the above it can be speculated that the reorientation of the Cr-doped (Bi,Sb)₂Te₃ seems to be a two-step process that is spread over a range of magnetic fields (approx. 0.1-0.2 T). Meanwhile, in the other sample, Cr reorientation is attributed to a very sharp conductivity change taking place in the narrow range of magnetic fields. I miss a comment clarifying the reason behind this.

Response: As discussed above, the Cr transition in the Cr-doped (Bi,Sb)₂Te₃ always occurs in a single step. The referee further raises an interesting observation of the apparent broadening in the magnetic transition of the Cr layer in Cr-doped (Bi,Sb)₂Te₃ as compared with the same Cr layer in the Cr-doped (Bi,Sb)₂Te₃/(Bi,Sb)₂Te₃/V-doped (Bi,Sb)₂Te₃ sample. Despite the appearance of the transport data, the width of the magnetic reorientation of the Cr layer in each case is roughly equivalent. The illusion of broadening in the CBST film is due to complications of evaluating magnetic behavior indirectly through the transport which is an electronic probe. Particularly, the magnetic reorientation in CBST appears broader, because this switching drives an electronic transition between two QAHI phases (with Chern numbers of +/-1), while in the case of the trilayer structure Cr layer reversals drive transitions between QAHI and AXI states (C = 1 and C = 0). Consequently, the signature in the conductivity measurement is fundamentally different.

Figure R3. Magnetization reversal in the Cr-doped $(\text{Bi,Sb})_2\text{Te}_3$ and Cr-doped $(\text{Bi,Sb})_2\text{Te}_3/(\text{Bi,Sb})_2\text{Te}_3/\text{V-doped } (\text{Bi,Sb})_2\text{Te}_3$. Reproduced from Fig. 1 in the main text.

To understand this, we refer to Fig. 1 from the main text, reproduced above. In the case of the Cr-doped $(\text{Bi,Sb})_2\text{Te}_3$ film, the entire Cr magnetic reversal, i.e. (i) to (iii) in Fig. R3a, is visible in the transport measurement as two peaks in σ_{xx} and one transition from $-h/e^2$ to h/e^2 in σ_{xy} . Meanwhile, in the trilayer sample, only half of the Cr reversal can be resolved in the transport experiment, i.e., the region between (i) and (ii) in Fig. 1d. The system is in a highly insulating state with longitudinal resistances in the MOhm range between (ii) and (iii), such that σ_{xx} vanishes, and σ_{xy} plateaus. Thus, although the Cr magnetization is still switching between (ii) and (iii), it does not produce any resolvable feature in the transport.

Although this means we cannot compare the full transition width in both samples, we can still compare the first half of the Cr layer magnetic transition, which produces a single peak in each sample. This comparison is presented in Fig. R4, demonstrating that the width of each is roughly equivalent.

Figure R4. Comparison of the σ_{xx} peak for Cr transition in Cr-doped $(\text{Bi, Sb})_2\text{Te}_3/(\text{Bi, Sb})_2\text{Te}_3/\text{V-doped } (\text{Bi, Sb})_2\text{Te}_3$ heterostructure (blue curve) and that in Cr-doped $(\text{Bi, Sb})_2\text{Te}_3$ (red curve). The field sweeping direction is rightward/leftward in **a/b**. The peaks are centered at zero field for comparison.

As one final point, we would like to note that while the width of these peaks does indeed reflect the width of the magnetic transition to some extent, these widths observed in electrical transport measurement, according to the finite-size scaling theory, are also determined by the inelastic scattering length through the scaling relation. Therefore, the breadth of these peaks should not be viewed as the true breadth of the magnetic transition, but rather a convolution of the magnetic switching field distribution and electronic coherence lengths.

3) Moreover, is it possible that Cr form inclusions or any kind of phase separation? Authors say that the layers are „uniformly doped” but can this be supported with any kind of post-growth characterization technique e.g. transmission electron microscopy (TEM)?

Response: We thank the reviewer for this comment. Though we do not have TEM from the specific wafer used for this current study, we have regularly performed TEM on Cr-doped $(\text{Bi, Sb})_2\text{Te}_3$ films (some results are shown in Fig. R3). These results consistently confirm a single phase free of dopant inclusions/clusters. In addition to these STEM results, we also used RHEED to monitor sample quality during the growth process. The RHEED pattern of a Cr-doped sample used in the present study is displayed in Fig. R3e. During the entire growth process, the RHEED pattern does not show any sign of phase separation.

Figure R5. **a-d**, TEM images of Cr-doped $(\text{Bi,Sb})_2\text{Te}_3$ film (adapted from Nano Lett. 22, 5735 (2022), Phys. Rev. Lett. 113, 137201 (2014), Nano Lett. 7, 9205 (2013), and Adv. Mater. 2207622 (2022), respectively). **e**, RHEED images of Cr-doped $(\text{Bi,Sb})_2\text{Te}_3$ film.

4) The coercive field dependence on the temperature is carefully accounted for. How about the temperature dependence of the range of fields over which the magnetic reorientation is spanned („width” of the reorientation in the magnetic field scale)? I miss a discussion about that and it seems that it can affect the values presented in Fig. 5b.

Response: In QAH materials the localization-delocalization transition is driven by the exchange gap m , and, by extension, the magnetization of the sample. Therefore, the referee is correct that the width of the magnetization switching is indeed expected to impact $\partial\sigma_{xy}/\partial B$. The temperature dependence of dM/dB during magnetic reversal events, however, is anticipated to be quite weak in the present materials, and consequently, has been omitted from the present analysis.

To illustrate the virtual temperature independence of dM/dB , we refer the referee to magneto-optical Kerr effect (MOKE) measurements performed on a 6 QL Cr-doped $(\text{Bi,Sb})_2\text{Te}_3$ film at temperatures between 10 K and 1.8 K. Although the magnetic coercive field grows by roughly a factor of 4.5 in this temperature range, the breadth of the transition is virtually constant. Since this temperature range is well below the Curie temperature, thermally activated magnetic fluctuations are already strongly suppressed, and the switching field distribution is dominated by sample inhomogeneities, the distribution of which is temperature independent. Therefore, it is anticipated that the relative temperature independence of dM/dB

extrapolates to the lower temperatures where scaling analysis is performed. This being said, we believe it would be of interest to further study the switching field distribution of these films at the dilution refrigerator conditions. Extracting accurate magnetic characterizations of thin films in dilution temperatures is extremely challenging, however, and outside of the scope of the present study.

Figure R6. MOKE result of a 6 QL Cr-doped $(\text{Bi,Sb})_2\text{Te}_3$ sample measured at different temperatures.

5) What are the Curie temperatures of the magnetic Cr and V doped layer? It seems that they shouldn't be close to the investigated region to rule out the magnetic critical behavior playing any role in the measurements.

Response: The referee is correct that the physics studied in this manuscript is only observable at temperatures well below the Curie temperatures of magnetic TI layers. In the case of these samples, the Curie temperature is roughly $20\text{ K} \sim 30\text{ K}$, such that all presented data are collected at temperatures of $0.05 T_c$ and below and the samples have well established long-range magnetic order. The Curie temperature of these films can be seen in the temperature dependent ρ_{xy} plots in Fig R7. Here, the sudden rapid increase in the transverse resistance marks the onset of magnetism and a nonzero anomalous Hall coefficient.

Figure R7. Temperature dependence of Hall resistance for a 6 QL V-doped $(\text{Bi, Sb})_2\text{Te}_3$, 6 QL Cr-doped $(\text{Bi, Sb})_2\text{Te}_3$, and a 3 QL V-doped $(\text{Bi, Sb})_2\text{Te}_3$ /6 QL $(\text{Bi, Sb})_2\text{Te}_3$ /3 QL Cr-doped $(\text{Bi, Sb})_2\text{Te}_3$, respectively.

6) After clarifying the origin of the separate transitions in Fig. 1b-c the similar discussion about the temperature dependence of the derivative of the transverse resistance on the magnetic field (Fig. 5b) for the Cr-doped only layer could be performed. This may add an important information to the discussion.

Response: We thank the reviewer for this comment regarding the power-law critical behavior in the QAH-NI transition. As shown in Fig. 1b-c, there are four QAH-NI transitions. Unlike in the QAH-AXI transition case where each transition is associated with the magnetization reversal in different layers, in the Cr-doped $(\text{Bi, Sb})_2\text{Te}_3$, all transitions are related to the reversal of the same magnetic layer. Naturally, the critical exponent for these transitions would have the same value.

Before applying scaling analysis to the separate transitions, we would like to note that, as we stated in the main text, discrete jumps that appeared in the two inner transitions make these transitions not suitable for scaling analysis (Fig. R5a). Therefore, we only focus on the transitions at the higher fields, and the results are displayed in Fig. R5b. As can be seen, critical power-law temperature dependences with consistent values of the critical exponents are revealed in both transitions.

Figure R5. **a**, Field dependence of σ_{xy} for the 6 QL Cr-doped $(\text{Bi, Sb})_2\text{Te}_3$. Four QAH-NI transitions (black arrows) are displayed in the hysteresis loop. The discrete jumps in the two inner transitions are highlighted by magenta arrows. **b**, Temperature dependent $(\partial\sigma_{xy}/\partial B)_{max}$ for the Cr left (2) and Cr right (2) transitions.

In the revised manuscript, we have added the above result in Supplementary Materials as Supplementary Fig. 7 (see #11 in the summary of changes).

7) Data presentation in Fig. 5b is inappropriate - the span of the Y-axis should be adjusted to the data. Due to the doubts and missing information as presented above I do not perceive the manuscript as

suitable for Nature Communications.

Response: Following the reviewer's suggestion, we have replotted Fig. 5b in the updated version (see #8 in the summary of changes).

Reviewer #3:

In the manuscript "Quantized resistance revealed at the criticality of the quantum anomalous Hall phase transitions", by Peng Deng et al., critical behavior of topological phase transitions, in particular the quantum anomalous Hall to normal insulator (QAH-NI), and quantum anomalous Hall to axion insulator (QAH-AXI) transitions are investigated. Through their new data analysis protocol, the critical behavior of the QAH-NI and QAH-AXI transitions are evaluated over a wide range of temperature and magnetic field. One of the main outcomes of their analysis is the observation that critical behaviors in quantum anomalous Hall transitions is due to electronic rather than magnetic origin.

The manuscript is clear and well written; and also, the scope of this work fits well in the frame of *Nature communications*. It will attract the interests of readers in the physical sciences community. I have few comments that need to be clarified before its final acceptance:

Response: We thank the reviewer for the nice summary and are very grateful to the reviewer for appreciating the scientific merit of our work. Below please find the point-to-point response to the reviewer's comments.

1. As far as I can read in the main-text and methods, the Cr- and V-doping levels in their heterostructures is not indicated? To which extend the Cr- and V-doping level would influence the critical behaviors of the QAH-NI and QAH-AXI transitions?

Response: The Cr-doping level is about 12% in the uniformly doped Cr-doped $(\text{Bi, Sb})_2\text{Te}_3$. In the Cr-doped $(\text{Bi, Sb})_2\text{Te}_3/(\text{Bi, Sb})_2\text{Te}_3/\text{V-doped } (\text{Bi, Sb})_2\text{Te}_3$ heterostructure, the Cr- and V-doping levels are about 15% and 10%, respectively. The doping levels are estimated by the growth rate obtained from RHEED oscillation, and are in line with our previous studies and studies from other groups [Phys. Rev. Lett. 113, 137201 (2014); Nat. Mater. 14, 473 (2015)]. In light of the reviewer's comment, we have added the corresponding information in the Methods (see #5 in the summary of changes below).

As for the question regarding the extent to which the magnetic doping level would influence the

critical behaviors of the QAH-NI and QAH-AXI transitions, we believe the doping level (in a reasonable range) should likely not impact the critical behaviors in the QAHI-NI and QAHI-AXI transitions. The critical exponents are dictated not by the microscopic details of the system, but rather by global symmetries, dimensionality, interaction length scales, etc. Given that consistent results are obtained in two types of QAH material systems with distinct magnetic structures, we believe it is unlikely that varying dopant concentration would impact any of these features. We concede, however, it is possible that dopant disorders might impact interaction length scales and thereby influence the QAHI critical exponents, as has been previously reported in quantum Hall materials.

Unfortunately, while it would likely be interesting to test the dopant impact on QAHI criticality, in practice it is very challenging to vary the doping level of Cr and V significantly without destroying the quantization of the system. Magnetic doping breaks the quantization in two ways. First, the magnetic dopants reduce the strength of spin-orbit coupling, driving the system towards a topologically trivial regime. Second, magnetic doping also repositions the Fermi level, and the quantization is realized only when the Fermi level lies within the magnetic gap. Therefore, only a very narrow range of dopant levels may be traversed while still preserving QAH physics.

2. Given the strong statement made that “Together, these findings provide repeated evidence that transport signatures at criticality in QAH insulators are robust against material variations.....” I have a question regarding “reproducibility” of these data: it is not clarified in the manuscript how many Hall bars/samples were measured for each particular types of two distinct material platforms illustrated in Fig. 1(a) and Fig.1(d). If more than one samples/Hall bar devices were measured and analyzed using the proposed data analysis protocol, authors should make clarifications on this, and it would be good to have this included in the manuscript. This will strengthen their claim and also help readers to understand how robust their new analysis protocol is.

Response: We thank the reviewer for this important comment. To show the reproducibility for both material platforms to realize the QAH-NI and QAH-AXI transitions, we performed additional measurements within the same sample as well as among different samples. First, we examined different transitions (occurred under opposite field sweeping directions) in the same sample, the results are shown in Supplementary Fig. 1 (QAH-NI transition) and Supplementary Fig. 2 (QAH-AXI transition). Second, we also performed additional measurements to examine the critical behaviors of the phase transitions in

different samples for both material platforms, the results are shown in Fig. R6-R9. In total, we have examined 12 transitions across 6 different samples (3 Cr-doped $(\text{Bi, Sb})_2\text{Te}_3$ samples and 3 Cr-doped $(\text{Bi, Sb})_2\text{Te}_3/(\text{Bi, Sb})_2\text{Te}_3/\text{V-doped } (\text{Bi, Sb})_2\text{Te}_3$ samples). All of these transitions show quantized values of the transport coefficients at the critical point, confirming the reproducibility of our results and the robustness of the new analysis protocol.

In the updated Supplementary Materials, Figs. R6 and R7 have been added (as Supplementary Fig. 3 and Fig. 4, respectively) to present the additional results for QAH-NI transitions in other Cr-doped $(\text{Bi, Sb})_2\text{Te}_3$ samples. Similarly, Figs. R8 and R9 have been included (as Supplementary Fig. 5 and Fig. 6, respectively) to present the additional results for QAH-AXI transitions in other Cr-doped $(\text{Bi, Sb})_2\text{Te}_3/(\text{Bi, Sb})_2\text{Te}_3/\text{V-doped } (\text{Bi, Sb})_2\text{Te}_3$ samples (see #9 and #10 in the summary of changes).

Figure R6. QAH-NI transition in Cr-doped $(\text{Bi, Sb})_2\text{Te}_3$ sample #2. **a**, B^* ($= B - \mu_0 H_c$) dependences of ρ_{xx} at different temperatures. **b**, B^* ($= B - \mu_0 H_c$) dependences of σ_{xy} at different temperatures.

Figure R7. QAH-NI transition in Cr-doped $(\text{Bi, Sb})_2\text{Te}_3$ sample #3. **a**, B^* ($= B - \mu_0 H_c$) dependences of ρ_{xx} at different temperatures. **b**, B^* ($= B - \mu_0 H_c$) dependences of σ_{xy} at different temperatures.

Figure R8. QAH-NI transition in Cr-doped $(\text{Bi, Sb})_2\text{Te}_3/(\text{Bi, Sb})_2\text{Te}_3/$ V-doped $(\text{Bi, Sb})_2\text{Te}_3$ sample #2. **a-c**, B^{V*} ($= B - \mu_0 H_c^V$) dependence of σ_{xx} , ρ_{xx} , and σ_{xy} , respectively, under different temperatures. **d-f**, B^{C*} ($= B - \mu_0 H_c^C$) dependence of σ_{xx} , ρ_{xx} , and σ_{xy} for the same sample.

Figure R9. QAH-NI transition in Cr-doped $(\text{Bi, Sb})_2\text{Te}_3/(\text{Bi, Sb})_2\text{Te}_3/$ V-doped $(\text{Bi, Sb})_2\text{Te}_3$ sample #2. **a-c**, B^{V*} ($= B - \mu_0 H_c^V$) dependence of σ_{xx} , ρ_{xx} , and σ_{xy} , respectively, under different temperatures. **d-f**, B^{C*}

($= B - \mu_0 H_c^C$) dependence of σ_{xx} , ρ_{xx} , and σ_{xy} for the same sample.

3. It would be good to clarify at which temperatures the spectra in Fig. 1(b)-(f) were acquired. I understand it is probably at 100 mK for the blue and 1000 mK for the red one?

Response: The data in Fig. 1(b)-(f) were all acquired at 100 mK. The blue and red curves were acquired with the magnetic field sweeping to the left and to the right, respectively. In light of the reviewer's comment, we have added the above information in the figure captions to improve the readability of the manuscript. (see #6 and #7 in the summary of changes)

-----Summary of changes-----

(All changes in the manuscript are highlighted in blue)

1. Line 20 page 4, we added the sentence “(see more discussion in Supplementary Information).”
2. Line 10 page 5, we rewrote the sentence below:
“Particularly, we will analyze two observable effects predicted by finite-size scaling theory $\sigma_{\alpha\beta}(\rho_{\alpha\beta}) = f[(L_s/\xi)^{1/\nu}] = f[(x - x_c)T^{-p/2\nu}]$.”
3. Line 12 page 5, we rewrote the sentence below and change the reference within:
“Although several experimental and theoretical works report critical resistance values equal to the h/e^2 in quantum Hall systems, whether this value is universal or bears larger significance remains a subject of debate.”
4. Line 21 page 8, we rewrote the sentences below:
“The critical exponent κ extracted from these experiments is nominally expected to be universal, and any changes in its values require a substantial material modification. On the other hand, reported values of κ in QAH systems exhibit significant sample-to-sample variations, which may stem from the presence of disorders that interact over different length scales as has been previously reported in quantum Hall systems. This introduces an additional uncontrolled parameter that complicates the comparison of critical behaviors among different samples with distinct magnetic structures, making it difficult to distinguish if the magnetic details of a given system exert any influence over the observed κ value. Fortunately, AXI materials, which feature two unique magnetic structures embedded in a

single system, provide an excellent testbed to extract the influence the magnetism exerts, if any, on the critical exponents.”

5. Line 4 page 10, we added the following sentences:

“For the Cr-doped $(\text{Bi, Sb})_2\text{Te}_3$ sample, the Cr doping level is $\sim 12\%$. In the Cr-doped $(\text{Bi, Sb})_2\text{Te}_3/(\text{Bi, Sb})_2\text{Te}_3/\text{V-doped } (\text{Bi, Sb})_2\text{Te}_3$ heterostructure, the Cr- and V-doping levels are about 15% and 10% , respectively. In both types of QAH samples, the Bi/Sb flux ratio is adjusted to tune the Fermi level into the magnetization gap to achieve quantization.”

6. Line 15 page 10, we rewrote the figure captions of Fig. 1c and 1d as:

“**b,c**, Field dependences of σ_{xx} and σ_{xy} respectively for a 6 QL Cr-doped $(\text{Bi, Sb})_2\text{Te}_3$ measured at 100 mK. The red and blue curves were acquired with the magnetic field sweeping to the left and to the right, respectively.”

7. Line 4 page 11, we rewrote the figure captions of Fig. 1e and 1f as:

“**e,f**, Field dependences of σ_{xx} and σ_{xy} respectively for a 3 QL Cr-doped $(\text{Bi, Sb})_2\text{Te}_3/6$ QL $(\text{Bi, Sb})_2\text{Te}_3/3$ QL V-doped $(\text{Bi, Sb})_2\text{Te}_3$ measured at 100 mK. The red and blue curves were acquired with the magnetic field sweeping to the left and to the right, respectively.”

8. We adjusted the scale of Fig. 5b.

9. In Supplementary Information, we added Supplementary Figs. 3 and 4 to show the results of QAH-NI transitions in other samples.

10. In Supplementary Information, we added Supplementary Figs. 5 and 6 to show the results of QAH-AXI transitions in other samples.

11. In Supplementary Information, we added Supplementary Fig. 7 to show the results of the power-law scaling behaviors in different QAH-NI transitions in Cr-doped $(\text{Bi, Sb})_2\text{Te}_3$.

12. In Supplementary Information, we added Supplementary Fig. 8 and the discussion of origin of the zero Hall plateaus in two QAH material systems.

REVIEWERS' COMMENTS

Reviewer #1 (Remarks to the Author):

I am perfectly satisfied with the authors' response to my comments on the original version of the manuscript and with the modifications made to the manuscript. I recommend publication of the article as it is now.

Reviewer #2 (Remarks to the Author):

With respect to the original submission, the authors have provided important information in the supplementary part, which addresses and clarifies all the remarks that had been raised during the first revision round. It allows me to recommend this interesting manuscript for publication.

Reviewer #3 (Remarks to the Author):

In the revised version of the manuscript, the authors have addressed all my concerns . I recommend the revised manuscript for publication in Nature Communications.